# TIME SERIES PREDICTION WITH EVENTS BASED CAUSAL REPRESENTATION LEARNING

## ABSTRACT

The value of time series prediction is getting more and more attention, and the prediction of time series data under event disturbance has been difficult, the different distribution of data before and after the event and the different distribution of dataset will lead to the poor prediction accuracy, robustness and generalisation ability of prediction model(CRP). In this paper, based on the causal representation learning, we design the SCM structure under event disturbance and propose the causal representation prediction model, which is divided into two parts, CRP_Encoder and CRP_Decoder. CRP_Encoder completes the extraction of causal representations disturbed by events and those not disturbed by events through the causal factor extractor and the causal representation decoupler; in order to learn the causal mechanism, the equivalence of conditional structure and causal mechanism is proved, and CNN network and causal representation coupler are designed in CRP_Decoder to learn casual representation and predict. The experimental results show that the CRP model has high prediction accuracy, good robustness and strong generalisation ability.

## 1 INTRODUCTION

How to accurately enhance prediction accuracy, robustness, and generalization of time series data under events disurbance is becoming important Kattan et al. (2015); Tzeng et al. (2014); **?**); **?**. Conventional statistical models or machine learning models, which fundamentally rely on the assumption of independent and identically distributed (IID)Annamalai et al. (2022); **?**); Lv et al. (2022b), can't deal well with out of distributed problem(OOD),like robustness, generalization. So, reaschers try to use causal mechanisms which are remain consistent across different datasets Lv et al. (2022b); Schölkopf et al. (2021) and similar event disturbance Bottou et al. (2013); Parascandolo et al. (2018) to solve this problem. Besides, some studies believes that events disturb the causal mechanisms between data before and after disturbance, leading to changes in data distribution which also belongs to out of distribution Schölkopf et al. (2021); Aghion et al. (2009). Thus, in this paper we use casual representation learning to extract causal representations and learn causal mechanisms,

This paper introduces the Causal Representation Prediction Model (CRP model), distinct from prior causal representation learning models. This model extracts causal representations that differentiate between those affected by an event and those unaffected. The paper outlines three essential properties for causal representations and designs the CRP_Encoder component accordingly. The CRP_Encoder comprises a causal factor extractor $g$ and a causal representation decoupler $G$. $g$ extracts causal representations from the data and employs $G$ to distinguish two different causal representations. To predict data changes by using the corresponding causal mechanisms linked to event-affected and event-unaffected causal representations, we introduce a CRP_Decoder. This decoder includes a Casual Catch Network (CCN), an event-independent causal representation prediction network $f$, and a causal representation coupler $h$. Leveraging proposed causal mechanisms and conditional structural equivalences, the CRP model improves upon conventional neural networks for causal representation learning to achieve accurate predictions. Contributions of this paper in comparison to prior work include:

- Extract and distinguish the casual represnetations and prove the causal mechanism is equivalent to the conditional structure if casual representations are given.

- Propose CRP model based on causal representation learning : CRP_Encoder can distinguish and extract the different representations; CRP_Decoder learns the causal mechanisms repectivly, makes the model highly highly prediction accuracy,robust and generalisation.

- The experimental results on the two datasets show that the CPR model outperforms other models in the three indicators, which verifies the prediction ability, robustness, and generalisation of the CRP model.

## 2 RELATED WORK

### 2.1 TIME PREDICTION METHODS UNDER EVENT DISTURBANCE

Time prediction tasks under event disturbance involve the recognition of casual relationship between events and data. The work by Pearl and others has provided a theoretical foundation for modeling causality Pearl (2009). Granger causality tests Granger (1988) are widely used methods for determining causality between two time series and have been applied extensively in time prediction tasks under event disturbance. Traditional statistical methods such as Autoregressive Integrated Moving Average (ARIMA) Shumway et al. (2017) often exhibit good fitting capabilities for both linear and nonlinear relationships but have limitations in capturing time structural changes induced by events.In addition, models based on neural networks have made significant progress in time prediction tasks under event disturbance. Models that incorporate event information, such as Event-LSTM Annamalai et al. (2022) and N-beats Oreshkin et al. (2019), have shown improved performance by increasing their dependency on event information, allowing for more accurate predictions after event disturbance. However, these models are still constrained by the Independent and Identically Distributed (IID) assumption, which limits their ability to address Out-of-Distribution (OOD) problems caused by events and structural changes.To tackle these challenges, recent research has shifted its focus towards approaches rooted in causal inference and transfer learning. For instance, in a study by Athey & Imbens (2016), causal inference was incorporated into decision tree models through recursive partitioning, effectively reducing the model's dependency on the IID assumption. In the realm of transfer learning, domain adaptation techniques, as proposed by Ganin & Lempitsky (2015) using Domain Adversarial Neural Networks (DANN), have been introduced to enhance model generalization by minimizing distribution discrepancies between source and target domains.

### 2.2 CAUSAL REPRESENTATION LEARNING

To harness the advantages of causal inference within the realm of machine learning, the concept of causal representation learning was introduced Ahuja et al. (2022). This innovation, as exemplified by Lv et al. (2022a), was initially applied to image classification tasks. It involved the identification of invariant causal mechanisms in image classification, resulting in remarkable classification accuracy. Moreover, Rebuffi et al. (2017) introduced the incremental classifier and representation learning (iCaRL) technique, which bolstered the resilience and generalization capabilities of machine learning systems.The iCITRIS method proposed by Lippe et al. (2022), a form of causal representation learning rooted in instantaneous effects, specializes in identifying causal factors from time serise data and constructing causal graphs through differentiable causal discovery approaches. Furthermore, Shen et al. (2022) introduced the disentangled generative causal representation (DEAR) learning method, which, when supervised information is appropriately applied, enables both causal controllable generation and causal representation learning.

While causal representation learning has been widely utilized to address out-of-distribution (OOD) challenges, it is worth noting that there is currently no research exploring the application of causal representation learning to tackle OOD problems in the domain of time-series prediction under event disturbance.

## 3 THEORY AND METHOD

In this paper, we consider the problem of time-series prediction under event disturbances from the perspective of causal representation learning, and develop a generic structural causal model for time-series prediction under event disturbances, as shown in Fig. 1.

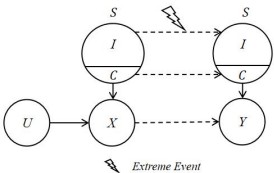

**Fig. 1.** SCM structure.

## 3.1 THEORETICAL DEMONSTRATION

**Principle 1** Penrose & Percival (1962): Causal factors $S$ contain information that explains all the dependencies of $X \rightarrow Y$.

Based on Principle 1, this paper formalizes the SCM under event as follows:

$$X := g(S(C\&I), U, V_1), C \perp I \perp U \perp V_1 \tag{1}$$

$$Y := h(do(f(C), do(I), V_2)), V_1 \perp V_2 \tag{2}$$

In Eq. (1) and Eq. (2), we denote $X$ and $Y$ as data before and after the disturbances. The causal factor $S$ has a mapping relationship with both $X$ and $Y$, while the non-causal factor $U$ only has a relationship with $X$. $S$ consists of event-related causal representation $I$ and event-unrelated causal representation $C$. $I$ contains information relevant to $Y$ and the corresponding causal mechanism influenced by event, while $C$ contains information relevant to $Y$ but with causal mechanism not influenced by event. $V_1$ and $V_2$ are unexplained noise variables. Model of event influence is represented by $do()$, $g$ and $f$ and $h$ can be considered as unknown models.

As above, $I$ contains information relevant to $Y$, so we use $p_{IY}$ to denote the probability of $I$ leading to $Y$ when $X$ as the condition space. It should be emphasized that $p_{IY}$ is not the same as $P(Y|I)$, as the former represents the probability of $I$ leading to $Y$ when the condition is $X$ (where $\neg I = C \cup U \cup V_1$), whereas the latter represents the probability of $I$ leading to $Y$ when the condition space is only $I$ (where $\neg I = \emptyset$). Similarly, we also define $p_{CY}$ as the probability of $C$ leading to $Y$ when $X$ the condition space. $p_{IY}$ and $p_{CY}$ are the causal mechanism related to event and unrelated to event. Based on this, as described by the SCM structure, the Bayesian distribution of $P(Y)$ can be obtained as:

$$P(Y) = P(I) \times p_{IY} + P(C) \times p_{CY} - P(I) \times P(C) \times p_{IY} \times p_{CY} \tag{3}$$

When the input of the model are $I$ and $C$, the statistical dependence learned by the general machine learning model can be expressed by $P(Y|I, C)$ Koller & Friedman (2009).

$$P(Y|I, C) = \frac{P(I \cap C) \times p_{IY} + P(C \cap I) \times p_{CY} - P(I \cap C \cap I \cap C) \times p_{IY} \times p_{CY}}{P(I \cap C)} \tag{4}$$

Because of $C$ is independent of $I$, $P(I \cap C) = P(C)P(I)$, Simplified Eq. (5):

$$P(Y|I, C) = p_{IY} + p_{CY} - p_{CY} \times p_{IY} \tag{5}$$

Since S contains all the information for the prediction Y, i.e.$p_{IY} + p_{CY} = 1$, thus, $p_{IY}, p_{CY} = \pm\sqrt{P(Y|I, C) - \frac{3}{4}} + \frac{1}{2}$ Based on the above discussion, if we can obtain the causal representation $I$ and $C$, it is easy to obtain the causal mechanism subject to event by optimizing the event model $h$ and get casual mechnisams $p_{IY}$ and $p_{CY}$

$$f*, do*, h* = \underset{h, do, f}{\arg\min} E_p[l(do(h(I), f(C), Y))] \tag{6}$$

where $l$ represents a certain loss function. Based on Eq. (6), if given causal representations $I$ and $C$, it is possible to optimise the model $h$ and learn the event-related causal mechanism $p_{IY}$ of $X \rightarrow Y$, and event-irrelated $p_{CY}$. Unfortunately, quite a number of researches Ahuja et al. (2022); Pearl (2019) have shown that the learning capability of existing machine learning models is limited, and relying on the optimisation alone it is very difficult to learn the causal mechanisms. In order to find a suitable model $h$ that learns causal mechanisms effectively, this paper will propose and prove

Principle 2, which states that causal mechanisms are equivalent to conditional structures given causal representations $I$ and $C$, in the expectation that causal mechanisms are learnt by machine learning models that learn conditional structures effectively Mirza & Osindero (2014); Scharstein & Pal (2007).

**Principle 2** van Rooij & Schulz (2019): The conditional structure $if\ a\ then b$ can be quantified using the value $\Delta^* p_a^b \frac{P(b|a)-P(b|\neg a)}{1-P(b|\neg a)}$. A higher value of $\Delta^{*b}a$ signifies stronger support for the given conditional structure. As for time prediction under the event disturbance, the existence of a conditional structure $if\ S then Y$ is assumed.

$$\Delta^* p_S^Y = \frac{P(Y|S) - P(Y|\neg S)}{1 - P(Y|\neg S)} \tag{7}$$

We assumes a variable $\Delta p_S^Y$ based on Vaswani et al. (2017), $\Delta p_S^Y = P(Y|S) - P(Y|\neg S)$, which represents the magnitude of the influence of $S$ within the space where $Y$ is generated. Based on Eq. (1) and Eq. (2), the causal factors represented and the non-causal factors represented, according to the Bayesian formula $P(Y) = P(U) \times p_{UY} + P(S) \times p_{SY} - P(U) \times p_{UY} \times P(S) \times p_{SY}$ and principle 1, $U$ does not contain information used for $Y$ prediction, i.e., $p_{UY} = 0$, it can be deduced that:$P(Y|S) = p_{SY} + P(U|S) \times p_{UY} - P(U|S) \times p_{SY} \times p_{UY}$, which can be obtained by transforming:

$$P(Y|\neg S) = \frac{[P(\neg S) \times p_{SY} + P(U \cap \neg S) \times p_{UY} - P(S \cap U \cap \neg S) \times p_{SY} \times p_{UY}]}{P(\neg S)} \tag{8}$$

where $P(U \cap \neg S) = P(U) \times P(\neg S)$, the proof is in the appendix,Therefore,

$$\Delta p_S^Y = 1 - P(U|S) \times p_{UY} \times p_{SY} + [P(U|S) - P(U|\neg S)] \times p_{UY} \tag{9}$$

From Eq. (9) we know:

$$p_{SY} = \frac{\Delta p_S^Y - [P(U|S) - P(U|\neg S)] \times p_{UY}}{1 - P(U|S) \times p_{UY}} \tag{10}$$

The derived result shows that $p_{SY} = \Delta^* p^Y S$ when $S$ is the cause of $Y$. In this scenario, the probability function $\Delta^{*Y} S$ for if $S$ then $Y$ is equal to $p_{SY}$. Hence, this study proves that the causal is equivalent to the conditional structure, given the causal representation $C$ and $I$.

Likewise, in the specified case of $S$, the knowledge of $\neg S$ value does not offer extra information for predicting $Y$, i.e, $P(Y|\neg S) \approx 0$, $P(Y|S) \approx 1$. Similarly, $p_{SY} = P(Y|S) = \Delta^* p_S^Y$, machine learning models that can effectively learn the conditional structure can be used to learn the causal mechanism.

As Bollen et al. (2011); Shojaie & Fox (2022), similar types events lead to comparable event in the data. As a result, general machine learning techniques can be employed to learn the event $do()$ data influenced by similar event. Thus, learning the causal mechanism between data before and after the event only necessitates having related causal representations. Regrettably, these causal representations are not given as priors. Directly reconstructing the causal representation and mechanisms is somewhat impractical since they are unobservable and ambiguous. However, causal representations must still meet specific principles. Peters et al. (2017); Schölkopf et al. (2012) suggested that causal factors should be jointly independent.

**Principle 3** Peters et al. (2017); Schölkopf et al. (2012): Independent Causal Mechanism (ICM), i.e., the causal mechanism of a given variable does not affect the causal mechanisms of other variables. The detailed descripitipn of Principle3 can be seen in appendix.

In summary, firstly, according to Principle 2, we believe that when the condition of the conditional structure the causal representation, the conditional structure is equivalent to the causal mechanism. Therefore, we can leverage the advantages of machine learning to learn the conditional and subsequently learn the causal mechanisms. Secondly, based on Principles 1 and 3, suggests that causal representations should be extracted by satisfying the following three fundamental properties:

- Property1: General neural network models learn that statistical dependence can distinguish between causal factors $S$ and non-causal factors $U$.
- Property 2:Changes in $I$ and $C$ respond to all changes between $X$ and $Y$.
- Property 3: Dimensions of causal representations should be jointly independent.

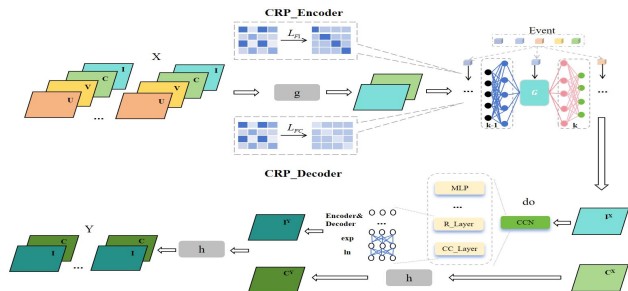

**Fig. 2.** CRP model structure.

## 3.2 CRP MODEL

The CRP model inspired by the above causal representation learning proposed in this paper consists of a CRP_Encoder composition and a CRP_Decoder component, as shown in Fig 2. where $do$ represents the disturbance generated by the target event and $k$ represents the number of the structural layer in the decoupler $G$. The whole CRP model is trained on data before and after the same type of event in history, the causal factor extractor $g$, the causal representation decoupler with event attention mechanism $G$ in the CRP_Encoder component, the prediction network $f$, and the causal factor coupler $h$. $g$ can be used to distinguish between the pre-event data with causal factors $S$ and non-causal factors $U$; $G$ represents the representation extraction structure with event attention mechanism, which is used to extract event-related causal representations $I$ from $S$; and $h$ represents the structure for learning causal mechanisms. $L_{FI}$ and $L_{FC}$ are the loss functions designed in this paper to optimise the extraction of $I$ and $C$ by $G$.

$$\begin{cases} S_\tau = g^*(X_\tau), g^* = \arg\min_g l(g(X_{\tau-1}, Y_{\tau-1}))I_\tau \\ I_\tau, C_\tau = G^*(S_\tau), G^* = \min_G L_{FC} \\ Y_\tau = h^*(do^*(I_\tau), f^*(C_\tau)), do^*, h^*, f^* = \arg\min_{do,h,f} E_p(l(h(do(I_{\tau-1}), f(C_{\tau-1}))), Y_{\tau-1}) \end{cases} \tag{11}$$

Where $l$ is an arbitrary loss function, $X_\tau$ is the target data, $X_{\tau-1}$ is the training data, by continuously inputting the training data $X_{\tau-1}$ and $Y_{\tau-1}$ to optimise formula (14), and finally get to extract the causal representation. Learning the causal mechanism of $g^*$, $G^*$ and $h^*$, complete the prediction of time series data $X_\tau$ under the disturbance of event $\tau$, as in Equation (14). $S_\tau$, $C_\tau$ and $I_\tau$ are the causal factor of the target data, the causal representation not related to the event, the causal representation related to the event, and $Y_\tau$ is the prediction result of $X_\tau$.

### 3.2.1 CRP_ENCODER

In order to be able to decouple the event-related causal representation $I$ and the event-independent causal representation $C$ from $X$, this paper designs the CRP_Encoder. the CRP_Encoder first extracts $S$ by satisfying property 1 through the causal factor extractor g, and then satisfies property 2 and property 3 of the causal representation through the causal representation decoupler $G$, the $L_{FC}$ and $L_{FI}$ loss functions, and then $I$ and $C$ are extracted from the $X$.

$$S_\tau = g^*(X_\tau), g^* = \arg\min_g l(g(X_{\tau-1}, Y_{\tau-1})) \tag{12}$$

$$C_\tau, I_\tau \& = G^*(S_\tau), G^* = \min_G L_{FC}, L_{FI} \tag{13}$$

The description and design of $g$ can be seen in the appendix.

This paper extracts the causal representations from $S$ by means of the causal representation decoupler $G$ with event attention mechanism, $L_{FC}$ and $L_{FI}$ loss functions. The causal representation decoupler implemented through the Transformer is denoted as $G$, with causal representation $I/C = G(S)$. In this paper, we hope that the decoupler $G$ can pay attention to the part related to the event which means the parameters of the event are input into the network weights for optimisation, so an event attention layer is set up between each layer of the Transformer. It is assumed that the

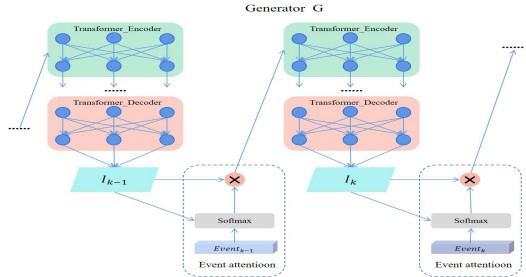

**Fig. 3.** CRP model structure.

events are all seen as loose events, consisting of a set of eigenvalues Chen & Li (2020). The disturbance of events $Event()$, m is the number of eigenvalues. The event attention layer is shown in Fig.3, where the value of the activation of the extracted representations from layer $k-1$th multiplied by the $k$th feature of the event is used as the attention weight, and the parameters of the $k$th layer of the representation decoupler $G$ are corrected to simulate the disturbance of the event's $k$th feature on the causal representation. The structural expression of Transformer with event attention mechanism is shown in Eq. (14) and (15).

$$I_k/C_k = G(Q = Event_kQ, K = Event_kK, V = V) \tag{14}$$

$$Event_k(I_{k-1}) = softmax(I_{k-1}/C_{k-1} \times E_k) \times I_{k-1}/C_{k-1} \tag{15}$$

$E_k$ represents the value of $k$th feature of the event, $I_k, C_k$ represents the representation extracted after the $k$th layer of transformer. When $k = m$, $I_k$ represents the disturbance-related causal representation $I$ that needs to be extracted ; when $k = 0$, $I_k$ represents the causality factor $S$ extracted by the time convolutional network. $Q$ is the query matrix in the transformer, $K$ represents the attention information related to the time sequence $Y$ after the event disturbance, and $V$ represents the abstract representation of $do_k(S)$ Vaswani et al. (2017). In order to ensure that $I$ and $C$ can be extracted and distinguished through $G$, further pairs of optimisations are proposed in this paper as in Eq. (16):

$$\max_G \frac{1}{J} \sum_{j=1}^{J} COR(c_j^x, c_j^y) \quad \min_G \frac{1}{J} \sum_{j=1}^{J} COR(i_j^x, i_j^y) \tag{16}$$

Extracting the representation of $X$ and $Y$ from the training set before and after the event gets $I$, denoted as $I^X$ and $I^Y$ respectively. $I^X = i_1^x, i_2^x, \ldots, i_J^x$ and $I^Y = i_1^y, i_2^y, \ldots, i_J^y$. Similarly, $C^X = c_1^x, c_2^x, \ldots, c_J^x$ and $I^Y = i_1^y, i_2^y, \ldots, i_J^y$. $i_j^x$ and $i_j^y$ represent the $j$th dimension of $I^X$ and $I^Y$ respectively, $c_j^x$ and $c_j^y$ represent the $j$th dimension of $C^X$ and $C^Y$ respectively, where $J$ is the dimension length. As in property 2, $I$ is more variable before and after the event, $C$ is unchanged before and after the event, therefore, the correlation between the values of $C$ in the same dimension before and after the event should be relatively large, whereas $I$ should be less correlated due to the influence of the event. To satisfy this condition, this paper uses $COR()$ to calculate the correlation and minimizes the correlation between the same dimensions $i_j^x$ and $i_j^y$ of $I$ and maximizes the correlation between the same dimensions $c_j^x$ and $c_j^y$ of $C$.

$$\min_G \frac{1}{J(J-1)} \sum_{j \neq k} COR(i_k^x, i_j^y) \quad \min_G \frac{1}{J(J-1)} \sum_{j \neq k} COR(c_k^x, c_j^y) \tag{17}$$

To unify the two optimization objectives mentioned above, this paper constructs a correlation matrix $C = \frac{<i_k^X, i_j^Y>}{||i_k^X|| \times ||i_j^Y||}(\frac{<c_k^X, c_j^Y>}{||c_k^X|| \times ||c_j^Y||})$. where, $k, j \in 1, 2, \ldots, J$ and $<>$ is inner product operation. According to Eq. (17), The correlations of different dimensions of $I, C$ need to be minimised; and according to Eq. (16), the correlations of the same dimensions of $I$ need to be maximised and the correlations of the same dimensions of $C$ need to be minimised. Based on this, the loss functions $L_{FC}$ and $L_{FI}$ are designed as in Eq. (18).

$$L_{FC} = \frac{1}{2}||QR\_SUM(COR) - 1||_F^2 \quad L_{FI} = \frac{1}{2}||QR\_SUM(COR) - 0||_F^2 \tag{18}$$

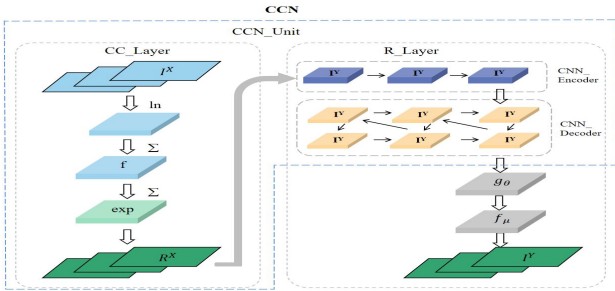

**Fig. 4.** CRP model structure.

$QR\_SUM(COR)$ represents the sum of eigenvalues after diagonalisation of the association matrix. According to Eq. (18) the sum of the diagonal elements is made as close as possible to 1 by wanting the sum of the diagonal elements to be as close to 1 when $C$ is extracted and as close to 0 when $I$ is extracted. According to Eq. (17), try to make non-diagonal elements of correlation matrix close to 0 by diagonalisation, i.e., the dimensions of causal representations are jointly independent. Thus, properties 2 and 3 of the event-related representations can be satisfied by minimising $L_{FC}$ and $L_{FI}$.

### 3.2.2 CRP_DECODER

In order to improve the accuracy, robustness and generalisation of timing prediction under event disturbance, the aim is to use the CRP_Decoder learn causal mechanism. The causal mechanism subject to event disturbance is further learnt by firstly learning the conditional structure of $I^x \to I^Y$ through the CCN in Decoder and introducing the effect of $do()$ of event disturbance as in Eq. (19).

$$I^Y_\tau = do^*(h^*(I^X_\tau)), do^*, h^* = mathop\arg\min_{do,h} E_p(l(do(h(I^X_{\tau-1}),)), I^Y_{\tau-1}) \qquad (19)$$

In order to be able to learn the conditional structure between $I^X$ and $I^Y$, CC_Layer is designed in the CCN component as in Eq. (20)

$$H(I^X) = \sum_{j=1}^{j=J} w_j exp(\sum_{i=1}^{i=K} p_{ij} ln(I^X)) \qquad (20)$$

$w$, $p$, and $b$ are unknown training parameters, $J$ and $K$ represent the number of parameters. The CC_Layer is a modified design based on the Product Unit Networks (PUNs) network. PUNs network is commonly used for discovering numerical patterns in various domains and is capable of effectively learning conditional structures like $if - then - else$ structures Zhang et al. (2021); Minaei-Bidgoli & Lajevardi (2008), which can be expressed as $Y = \sum_{j=1}^{j=J} w_j \prod_{i=1}^{i=K} (I)^{p_{ij}}$. However, the computational requirements are high when performing matrix operations and taking limits due to the influence of the multiplication calculations in the PUNs network. Additionally, multiplication by zero exceptions may occur. To address these issues, the CC_Layer optimizes the PUNs network by performing logarithmic calculations $\sum_{j=1}^{j=J} w_j exp(\sum_{i=1}^{i=K} p_{ij} ln(I))$, transforming the multiplicative factors into additive factors. This helps avoid multiplication by zero exceptions and mitigates the computational demand.

In addition to designing the CC_Layer to learn the conditional structure, this paper also introduces the R_Layer to incorporate the historical information of the event disturbance. The structure of the R_Layer is described by Eq. (21).

$$do(H(I^X)) = CCN_Decoder(b, WY, VCCN_Encoder(U \sum_{j=1}^{j=J} w_j exp(\sum_{i=1}^{i=K} p_{ij} ln(I^X))) \qquad (21)$$

Then to improve the predictive ability of the model, we add two layers $g_\theta$ and $f_\mu$. Details can be seen in appendix. Then, the CCN component as a whole consists of a multi-layer CCN_Unit and two linear mapping layers as in Eq. (22).

$$I^Y = do(H(I^X)) = f_\mu(\sum g_\theta(Decoder(b, WY, VEncoder(U \sum_{j=1}^{j=J} w_j exp(\sum_{i=K}^{i=K} p_{ij} ln(I^X)))))) \qquad (22)$$

Where, $V, U, W, w, p, b$ are all unknown training parameters, $f_\mu$ and $g_\theta$ are the mapping layers of the parameters and respectively, in this paper, a two layer MLP is used. The CCN component designed in this paper is able to efficiently learn the causal mechanism between $X$ and $Y$. This is due to the fact that the conditional structure between $X$ and $Y$ can be learnt through the CC_Layer, and the R_Layer is able to use the information from the disturbance of the same type of events as parameters with the conditional structure's outputs to be combined for causal mechanism learning. The description and design of $f$ can be seen in the appendix.

## 4 EXPERIMENT

The description of dataset and parameters can be seen in the appendix.

### 4.1 VERIFICATION EXPERIMENT

The prediction accuracy, robustness of the CRP model is experimentally compared with two common timing prediction methods, RNN and seq2seq, and with two newer proposed prediction methods, Dilate**?**, and N-beats, on the above two datasets. The results of the experimental comparison are shown in Fig.5, Table 1, and appendix. From the results of the MSE, MAE, and RMSE mea-

| Dataset | Dataset1 | | | Dataset2 | | |
|---|---|---|---|---|---|---|
| **Method** | **MSE** | **MAE** | **RMSE** | **MSE** | **MAE** | **RMSE** |
| CRP | **0.1010** | **0.2620** | **0.3178** | **0.0878** | **0.2495** | **0.2964** |
| RNN | 2.5059 | 1.3943 | 1.5830 | 2.4870 | 1.3932 | 1.5770 |
| Seq2seq | **0.6754** | **0.7206** | **0.8218** | **1.0647** | **0.7332** | **1.0318** |
| Dilate | 0.7733 | 0.7672 | 0.8793 | 1.0868 | 0.7891 | 1.0425 |
| N-beats | 2.4647 | 1.3921 | 1.5697 | 1.1486 | 1.0616 | 1.0717 |

Table 1: Results of the MSE, MAE, and RMSE measures on different datasets.

sures, the CRP model achieved better results on both datasets. Specifically, the MSE, MAE, and RMSE scores of the CRP model are 57.44%, 45.86%, and 50.40% higher than the corresponding scores of the Seq2seq model, which is the second highest, on dataset 1, and 97.69%, 48.37%, and 73.54% higher on dataset 2, respectively. Meanwhile, the differences of the three measures of the

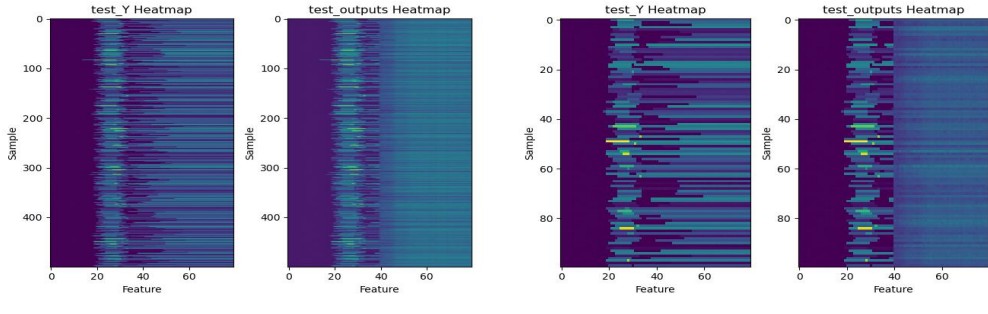

(a) Dataset1 1:All samples comparasion      (b) Dataset2:All samples comparasion

**Fig. 5.** Test results on different dataset.

CRP model are the smallest on both datasets, with the MSE score difference of 0.0132, which is the lowest among all the models, which indicates that the prediction accuracy of the CRP model fluctuates the least on different datasets, and has a good robustness.

### 4.1.1 EXPERIMENTS ON THE COUNTERFACTUAL PREDICTON OF CRP MODEL

If a model learns the causal mechanism of disturbed by events, then by changing only the disturbance of events, the model can also get the corresponding results, i.e., the model has the ability to answer

counterfactual questions. Based on this, in order to verify the ability of the CRP model to answer counterfactual questions, all the parameters of the dataset 2 events were taken as negative values, and a new test set was generated for the experiment, and the results of the experiment are shown in Table 2 and appendix. It can be found that the CRP model also has relatively high prediction accuracy

|  | MSE | MAE | RMSE |
|---|---|---|---|
| **Test_unfact** | 0.1722 | 0.3345 | 0.4150 |
| **Test** | **0.1010** | **0.2620** | **0.3178** |

Table 2: Results of the counterfactual experiment.

for the counterfactual situation, with an MSE score of 83.78%. Neither the event parameters taken in the counterfactual nor its corresponding pre-event data of the test set were present in the training set, yet the CRP model was able to predict the counterfactual outcome better as shown appendix, inferring that the CRP model has the ability to predict the counterfactual scenarios, i.e., the CRP model learns the causal mechanism and has a strong generalisation ability.

## 5 CONCLUSION

Aiming at the time series prediction under event disturbance, this paper proposes a structure of SCM under event disturbance based on causal representation learning, by obtaining the causal representation $I$ which is disturbed by the event and the causal representation $C$ which is not disturbed by the event and learning the corresponding causal mechanism to achieve the prediction of the data. The causal representation prediction model for time series data under event disturbance, CRP model, which consists of two parts, CRP_Encoder and CRP_Decoder. CRP_Encoder designs the causal factor extractor $g$, based on the property that the $g$ can distinguish between $S$ and $U$; because $I$ and $C$ in $S$ can react to the change between $X$ and $Y$, through the nature of causal representation dimensions are independent of each other, the causal representation decoupler $G$ is designed to achieve $X \rightarrow I, C$. Meanwhile, the equivalence rule of causal mechanism and conditional structure is proved, which enables the neural network to learn the causal mechanism efficiently. Under the premise of obtaining the causal representations, based on the equivalence rule, the CRP_Decoder's CCN is designed and causal representation coupler $h$ to learn the corresponding causal mechanism to achieve $I, C \rightarrow Y$.Finally, the prediction of $X \rightarrow Y$ is completed by CRP_Encoder extracting causal representation and CRP_Decoder learning causal mechanism.

Comparing the three experimental results of different models on the two datasets, the CRP model achieves the best scores, and gets up to about 89% of the MSE scores on Dataset 1, which indicates that the CRP model has a high prediction accuracy. Comparing the differences of all models for each experimental metric on the two datasets, the maximum difference of the CRP model is only 0.0132, which is the lowest, indicating that the CRP model has better robustness. Analysing the results of the counterfactual prediction experiments, the MSE score of the CRP model also reaches about 83%, indicating that the CRP model has good generalisation ability. Comprehensive analysis of the experimental results shows that the CRP model can effectively extract the causal representations that distinguish between those that are disturbed by events and those that are not disturbed by events in the data, and learn the corresponding causal mechanisms.

This paper extracts and distinguishes causal representations, learns the corresponding causal mechanisms, and improves the predictive, robustness and generalisation abilities of the model, it does not make assumptions about the noise, which may have an impact. This will be studied in depth in the future.

## 6 ACKNOWLEDGEMENTS

This work was supported in part by the National Key R&D Program of China under Grant No. 2020YFB1707900 and 2020YFB1711800; the National Natural Science Foundation of China under Grant No. 62262074, U2268204 and 62172061; the Science and Technology Project of Sichuan Province under Grant No. 2022YFG0159, 2022YFG0155, 2022YFG0157.

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

## A  APPENDIX

### A.1  THE PROOF OF $P(U \cap \neg S) = P(U) \times P(\neg S)$

Due to the independence between $U$ and $S$, $P(\neg U \cap \neg S) = P(\neg U) \times P(\neg S)$, it follows from De Morgan law that: $P(\neg(U \cup S) = P(\neg U) \times P(\neg S)$. Taking the negation on both sides yields $P(U \cup S) = 1 - P(\neg U) \times P(\neg S)$, i.e., $P(U) + P(S) - P(U \cap S) = 1 - (1 - P(U)) \times (1 - P(S))$. Simplifying further, we have $P(U \cap \neg S) = P(U) \times P(\neg S)$. Substituting $P(U \cap \neg S) = P(U) \times P(\neg S)$ into Eq. (9) yields $P(Y|\neg S) = P(U|\neg S) \times p_{UY}$.

## A.2  THE DISCRIPTION OF PRINCIPLE3

In general, $I$ can be decomposed into multiple as $I = i_1, i_2, \ldots, i_H$, and $p_{IY} = p^{i_1}IY, p^{i_2}IY, \ldots, p^{i_{H_I}}IY$, where $i_k$ represents the values corresponding to different dimensions, $p^{i_k}_{IY}$ represents the causal mechanism corresponding to $i_k$, and $H_I$ represents the number of dimensions in $I$. Building upon the ICM Principle, we extend the ICM Principle to causal representations. Firstly, the event mechanism $p^{i_k}Y$ does not affect any other mechanisms $p^{i_j}IY$, where $k \neq j$. Secondly, even if information about other disturbance-interfered mechanisms $p^{i_j}_{IY}$ is known, it does not provide information about the mechanism $p^{i_k}_{IY}$. Therefore, the final causal mechanisms under disturbance can be decomposed into $do(p_{IY}) = \prod_{k=1}^{H_I} do(p^{i_k}_{IY})$. Similarly, Representation $C$ can be decomposed and needs to satisfy independent causal mechanisms. The final causal mechanism that is not disturbed by events is decomposed into: $f(p_{CY}) = \prod_{k=1}^{H_C} f(p^{c_k}_{CY})$, that is, $c_k$ is the values corresponding to different dimensions after decomposition, $H_C$ and the dimension length of $C$.

## A.3  THE DISCRIPTION OF $g$

Although the display forms of the causal factor extractor $g$ and causal representation decoupler $G$ in Eqs. (1) and (2) proposed in 3.1 are unknown, according to the previous analyses in this paper, $U$ is free of information related to $Y$, and the discrepancy between $X$ and $Y$ due to $U$ can be resolved by learning statistical dependencies. In some articles, it is indeed shown that some machine learning methods can sift out information that is not relevant to the predicted outcome **??**, such as time Convolutional Neural Networks **?** (TCN Networks).TCN Networks efficiently extract information relevant to the predicted outcome by means of structures such as causal convolutional layers. Formally, given the pre-event time-series data $X$, it can be represented after the time convolutional network as: $S = g(X)$, where $g$ represents the causal factor extractor implemented with the TCN network. As mentioned in **?** causal convolution results in keeping the time series data causal in the time dimension. Therefore, it is believed in this paper that the TCN network is sufficient to distinguish the causal factor $S$, which is related to the post-event outcome, satisfying property 1, and that the difference between $S$ and $Y$ that still exists is due to the event-related causal representation $I$, which is disturbed by the event, and the event-independent causal representation $C$.

## A.4  THE DESCRIPTION OF THE TYPE OF CCN_ENCODER&CCN_DECODER&OTHER LAYER

Based on the fact that disturbance generated by the same type of events are similar, the event disturbance $do()$ can be learned from data subject to similar event disturbance through general machine learning techniques. Therefore, an CCN_Encoder is designed in R_Layer to encode the output $h(I^X)$ of CC_Layer. The training data $I^Y_{\tau-1}$ is then fed into the CCN_decoder as a parameter (i.e., the disturbance result of the same type of event) along with the encoding of $h(I^X)$. The CCN_decoder and CCN_encoder can be in various forms, and the LSTM is used as the decoder and encoder for training.The CC_Layer and R_Layer together form the CCN_Unit, and the learning of the causal mechanism of the event-disturbed causal mechanism between $I^X$ and $I^Y$ is realised through a multilayer CCN_Unit and a two-layer mapping, where the two mapping layers better capture the core common properties of the causal mechanism **?**.

## A.5  THE DESCRIPTION OF $f$

Since the causal mechanism corresponding to $C$ is not disturbed by events, it only needs to be a model that can cater for the fact that learning has changed slightly, as models such as the MLP can meet the requirements. Similarly, considering the causal mechanism $f_\tau(p_{CY} = \prod_{k=1}^{H_C} f_{\tau(p^{c_k}_Y)})$ which is not disturbed by events, the explicit form of PUN, which employs the above mentioned improvements, $f$ is shaped as in Eq. (29).

$$C^Y = f(C^X) = \sum_{j=1}^{j=J} w_j exp(\sum_{i=1}^{i=K} p_{ij} ln(C^X)) \tag{23}$$

Finally, through the causal factor coupler, $C$ and $I$ are coupled into the $Y$ that needs to be predicted. The explicit form of $h$ can likewise be in various forms, and considering the symmetry of decoupling

Table 3: The introduce of dataset1

|  | **Num** | **Event_num** | **Sub_series_length** | **AUC of subseries** |
|---|---|---|---|---|
| Sum | 20758 | 200 | / | 0.9900 |
| Train | 14531 | 140 | 20 | 0.9921 |
| Test | 6227 | 60 | 20 | 0.9852 |

and coupling Scharstein & Pal (2007) a general Transformer network is used as the causal factor coupler.

### A.6 PRETREATMENT

The data preprocessing in this paper consists of two primary steps: First, leveraging adversarial validation in combination with the hypothesis from **?** to filter out event and data before and after the event; Second, reducing the differences between the data through linear normalization. The results of the adversarial validation are displayed in appendix The results of the pre-processing experiments

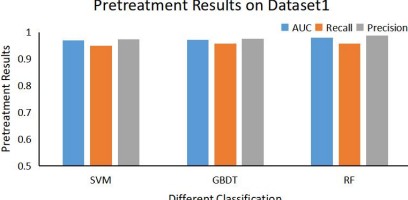 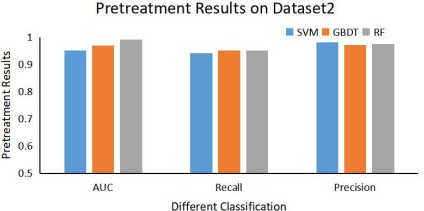

(a) Experiment 1:pretreatment on dataset1          (b) Experiment 2:pretreatment on dataset2

**Fig. 6.** Pretreatment on different dataset.

on the two datasets show that the AUC scores of the three classifiers are above 0.5, indicating that the distribution of the data before and after the event is inconsistent.

### A.7 EXPERIMENTAL DATASET

**Dataset 1** comprises daily voltage data (measured in voltage) recorded by sensors in an enterprise. Every 20 minutes, the are adjusted by workers, resulting in worker operations being the event in Dataset 1. More details about Dataset 1 can be found Table 1.

The training and test sets for Dataset 1 are obtained by dividing the pre-processed data in a 7:3 ratio. The AUC scores of the data before and after event are greater than 0.5, demonstrating that the pre-treatment data with notably different distributions before after event can be utilized to assess the predictive capability of the model under event. The KS_Score of the output from the training and test set is 0.9606, which is close to 1, indicating a distinct distribution of the training output and the test set output. Therefore, the training set and validation set can be employed to verify the robustness of the model.

**Dataset 2** is designed to assess the model's capability to address counterfactual questions. In this research, dataset 2 is generated using a random function based on the mean, variance, and other features of dataset 1. The generation process is described in Algorithm 1.

Table 4: The introduce of dataset2

|  | Num | Event_num | Sub_series_length | AUC of subseries |
|---|---|---|---|---|
| Sum | 40000 | 1000 | / | 0.9824 |
| Train | 20000 | 500 | 20 | 0.9726 |
| Test | 20000 | 500 | 20 | 0.9921 |

Table 5: Experiment parameters of G

| Method | Epoches | Hidden_size | d_model | nhead | num_layers |
|---|---|---|---|---|---|
| Transformer_Event | loss¡0.001 break out | 32 | 64 | 4 | 2 |

---

**Algorithm 3** Generated dataset Algorithm

---

**Input:** $N = 40000$, $\sigma^2 = 0.01$, $k = 20$
**Output:** Generated dataset

    Generate a dataset $D$ of size $N$ with random numbers in the range of 0 to 1, having a variance of $\sigma^2$
    Initialize an empty list $disturbance\_Points$
    $disturbance\_lengthk =$ random integer $\in [1, 20]$
    **for** $i = 1$ to $N$ with step $2 * k$ **do**
      Add $D_i$ to $disturbance\_Points$
      **for** $j = 0$ to $k - 1$ **do**
        **if** $i + j \leq N$ and $ij \geq 0$ **then**
          $D_{i+j} = D_{i+j} +$ random number
        **end if**
        Put $D_{i+j}$ into $subseries\_before$
        Put $D_{i-j}$ into $subseries\_after$
      **end for**
    **end for**
    **for** $i = 1$ to length of subseries **do**
      AUC=Adversarial validatioin(subseries_before, subseries_after)
      **if** $AUC \geq 0.5$ **then**
        Put $subseries\_before, subseries\_after$ into generated dataset
      **end if**
    **end for**
    Return $Generated dataset$

---

Algorithm 1 first creates a random sequence based on the input parameters; then it selects the time point at which the event occurs, and performs event disturbance on the data before and after the time point at which the event occurs, to obtain dataset 2, see Table 2. Dataset 2 is divided by 1:1 between training and test sets. The validity of the model was verified by decreasing the ratio of the training set to get more test sets. The lowest AUC score for the data before and after the event in dataset 2 is 0.9726 (greater than 0.5), and the KS_Score for the output of the training and test sets is 0.9502 (close to 1), so dataset 2 can also be used to validate the predictive ability and robustness of the model.

## B  EXPERIMENTAL PARAMETERS

The experimental environment and equipment configuration are as follows: operating system: Windows 10, processor: Intel Core(TM) i5-7300HQ 7CPU @ 2.50GHz, language: Python3.7, IDE: JetBrains PyCharm Community Edition. in order to help better understand and reproduce the experiments, some experimental parameters are provided in this paper.

Table 6: Experiment parameters of CCN.

| Method | Hidden_size | Num_Layers | Encoder(Decoder) Type | Loss Type |
|---|---|---|---|---|
| R_Layer | 128 | 2 | LSTM | Quantile_Loss |
| CC_Layer | 32 | 2 | / | Quantile_Loss |

### B.0.1 EXPERIMENTAL COMPARISON OF DIFFERENT TYPES OF CNN_ENCODER AND CNN_DECODER

The experimental comparison of the encoder and decoder of R_Layer using three network structures, RNN, LSTM, and GRU, shows that the difference in the final convergence values of the three structures is found to be small on dataset 1, and the fluctuation of the Smooth_L1 values of the test set obtained using the LSTM model as the encoder and decoder is also minimal as shown in Fig.7 . Therefore it is reasonable to adopt LSTM as the encoder and decoder of R_Layer

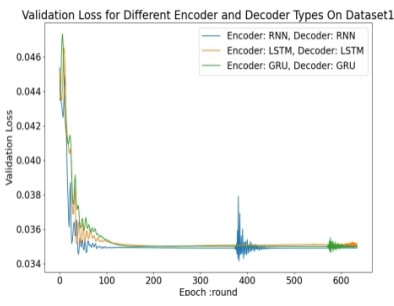

(a) Experiment 1:pretreatment on dataset1

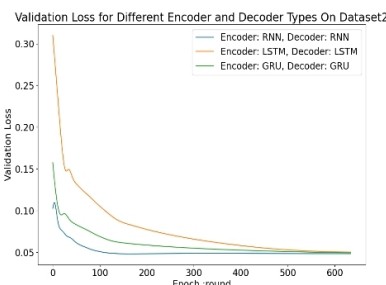

(b) Experiment 2:pretreatment on dataset2

**Fig. 7.** Pretreatment on different dataset.

### B.1 THE SPECIFIC RESULTS OF PREDICITON

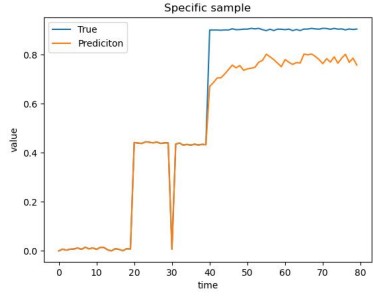
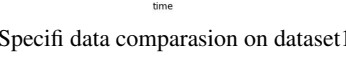

(a) Specifi data comparasion on dataset1

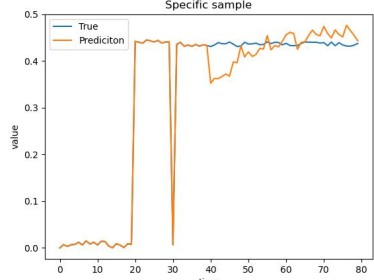

(b) Specifi data comparasion on dataset2

**Fig. 8.** Pretreatment on different dataset.

### B.2 COMPARASION OF UNFACT PREDICTION RESULT AND TEST RESULT

### B.3 DATASET AVAILABILITY

The datasets we used in our research are available at the following links:

- Dataset1:Due to confidentiality reasons, we are unable to provide the first dataset directly.

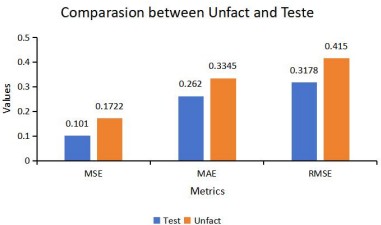

**Fig. 9.** SCM structure.

- Dataset2:
  https://www.kaggle.com/code/ekrembayar/store-sales-ts-forecasting-a-comprehensive-guide

You can get our source code from the link below:

- source code: `https://github.com/342869125/CCR_Model_Causaual`

If you need the complete source code, please contact the first author via email.

