# OpenReview forum: "Time Series Prediction With  Events Disturbance Based Causal Representation Learnin"
_ICLR.cc/2024/Conference — Submitted to ICLR 2024_

### Official Review · Reviewer_Huym · 2023-10-31

**Soundness:** 2 fair
**Presentation:** 1 poor
**Contribution:** 2 fair
**Rating:** 3
**Confidence:** 3

**Summary:**

This paper tackles time series prediction in the presence of event disturbances. It introduces a structured causal model tailored for scenarios involving such disturbances. The proposed causal representation prediction model contains two distinct components: "I," which is influenced by the event, and "C," which remains undisturbed and serves as the basis for prediction.

**Strengths:**

This paper considers an important problem---time series prediction under event disturbance. It uses a causal representation learning-based method to decompose the data into two components and utilize the invariant component for prediction.

**Weaknesses:**

The readability of this paper requires significant improvement, particularly in Section 3.1 where clarity is lacking. Despite the apparent importance of the SCM depicted in Figure 1 and detailed in Section 3.1, I found it challenging to follow.

Furthermore, it's worth noting that the inclusion of grant information in the submission raises concerns about violating the anonymous policy.

**Questions:**

The introductions to Figure 1, Principle 1, and the following SCMs should be clearly stated.

---

### Official Review · Reviewer_PJpw · 2023-10-31

**Soundness:** 2 fair
**Presentation:** 1 poor
**Contribution:** 2 fair
**Rating:** 3
**Confidence:** 5

**Summary:**

This work propose to use causal representation learning for time series prediction under event disturbance.

**Strengths:**

- The proposed CRP has higher accuracy compared with the other methods.

**Weaknesses:**

- This work seems far from complete and possibly violates the anonymous instruction due to the affiliation in the acknowledgment.
- The writing has many glitches. For example, what is the full name of CRP? IID => i.i.d. The unresolved citation.
- Many notations are introduced without definition.
- Some principles are discussed in section 3.1 but there is no theoretical guarantee on how to leverage the causal representation learning for the time series prediction and how to identify the corresponding causal representation.
- Many related works of causal representation learning are missing. The contribution of this work and the difference with the recent works should be stated clearly.

**Questions:**

See the weaknesses above.

---

### Official Review · Reviewer_arwZ · 2023-11-01

**Soundness:** 1 poor
**Presentation:** 1 poor
**Contribution:** 2 fair
**Rating:** 3
**Confidence:** 3

**Summary:**

Predicting time series under event disturbance can be difficult. This paper proposes an encoder to factor the causal representations into two parts: those which are disturbed by the events and those which are not. It also provides a decoder to make predictions.

**Strengths:**

The studied problem and the proposed model in this paper are interesting.

**Weaknesses:**

1. The paper is poorly written and hard to follow. There are tons of grammar mistakes. Even for the keyword "causal", the authors mistakenly used the word "casual" instead for 7 places.

2. There are no clear definitions of the mathematical notations, which makes it impossible to follow the meaning of the mathematical models. On a probably minor point, formulas are poorly presented in a chaotic way. For example, a formula should be interpreted as a part of the sentence, which means that there should be a comma or period after it if the sentence is to be finished.

3. The experimental results seem to be unreliable. The authors only select a few benchmark algorithms without explaining the reasons and without proper citations and conduct them on merely two datasets (no clear sources or citations again). As I understand, the second dataset is constructed by the authors based on the first one, implying that there may be only ONE underlying dataset for evaluation. There are even missing parts in the experiment results in the appendix. The paper should be taken more seriously to meet the caliber of a top conference.

**Questions:**

As above.

---

### Official Review · Reviewer_q7YY · 2023-11-09

**Soundness:** 2 fair
**Presentation:** 2 fair
**Contribution:** 2 fair
**Rating:** 3
**Confidence:** 4

**Summary:**

This paper focuses on the causal representation learning in the presence of event disturbance. This seems an interesting and realistic problem.
The proposed Causal Representation Prediction Model is an encoder-decoder structure.
The encoder is designed to extract causal representation disturbed by events and then the decoder learns to predict with the representations.
Some experiment results are provided to compare with some baselines.

**Strengths:**

This paper is focused on an interesting problem, i.e., learning causal representation for robust prediction in the face of event disturbance. However, the technical contribution and novelty are low.

**Weaknesses:**

1. The presentation issue makes the paper not ready for publishing, e.g., missing references.

2. The problem formulation is not in line with the claims in the introduction. The event disturbance is not explicitly and clearly defined.

3. Formula definitions and derivations are confusing, e.g., under Eq.(5), how is the equation derived?

4. The experiment section is poorly presented, e.g., no dataset and setup description, and the baselines are not representative and strong enough.

**Questions:**

See the comments above.

---

### Meta-Review · Area_Chair_kN4f · 2023-12-12

**Metareview:**

This paper is concerned with time series prediction under extreme event disturbances, which is a challenging problem. This paper proposes to address this problem based on causal representation learning. The idea may have some nice merits; however, the writing of the paper is to be improved in multiple ways to be published.

**Justification For Why Not Higher Score:**

The presentation of the paper needs a lot of improvement; for instance it contains a lot of missing references and many grammatical mistakes.

**Justification For Why Not Lower Score:**

The idea seems plausible.

---

### Decision · Program_Chairs · 2024-01-16

Reject